# COVID-19 Time of Intubation Mortality Evaluation (C-TIME): A system for predicting mortality of patients with COVID-19 pneumonia at the time they require mechanical ventilation

**Robert A. Raschke**[1,2]*, **Pooja Rangan**[2,3], **Sumit Agarwal**[2,3], **Suresh Uppalapu**[2,3], **Nehan Sher**[2], **Steven C. Curry**[1,4], **C. William Heise**[1,4]

**1** The Division of Clinical Data Analytics and Decision Support, University of Arizona College of Medicine-Phoenix, Phoenix, AZ, United States of America, **2** University of Arizona College of Medicine-Phoenix, Phoenix, AZ, United States of America, **3** Department of Internal Medicine, Banner—University Medical Center Phoenix, Phoenix, AZ, United States of America, **4** Department of Medical Toxicology, Banner–University Medical Center Phoenix, Phoenix, AZ, United States of America

* Raschkebob@gmail.com

**Data Availability Statement:** Data cannot be shared publicly due to stipulations of our data use

## Abstract

### Background

An accurate system to predict mortality in patients requiring intubation for COVID-19 could help to inform consent, frame family expectations and assist end-of-life decisions.

### Research objective

To develop and validate a mortality prediction system called C-TIME (COVID-19 Time of Intubation Mortality Evaluation) using variables available before intubation, determine its discriminant accuracy, and compare it to acute physiology and chronic health evaluation (APACHE IVa) and sequential organ failure assessment (SOFA).

### Methods

A retrospective cohort was set in 18 medical-surgical ICUs, enrolling consecutive adults, positive by SARS-CoV 2 RNA by reverse transcriptase polymerase chain reaction or positive rapid antigen test, and undergoing endotracheal intubation. All were followed until hospital discharge or death. The combined outcome was hospital mortality or terminal extubation with hospice discharge. Twenty-five clinical and laboratory variables available 48 hours prior to intubation were entered into multiple logistic regression (MLR) and the resulting model was used to predict mortality of validation cohort patients. Area under the receiver operating curve (AUROC) was calculated for C-TIME, APACHE IVa and SOFA.

agreement with Bannerhealth. Data are avaiable upon request from Kieran Richardson, Research Development Manager, University of Arizona College of Medicine-Phoenix (KieranR@arizona. edu).

**Funding:** This study was funded in part by the Flinn Foundation (grant 2196). The funders had no role in study design, data collection and analysis, decision to publish, or preparation of the manuscript.

**Competing interests:** The authors have declared that no competing interests exist.

**Abbreviations:** **APACHE**, acute physiology and chronic health evaluation; **AUROC**, area under the receiver-operating curve; **BMI**, body mass index; **COPD**, chronic obstructive pulmonary disease; **COVID-19**, coronavirus disease 2019; **C-TIME**, COVID-19 time of intubation mortality evaluation; **FiO2**, fraction of inspired oxygen; **MLR**, multiple logistic regression; **PaO2**, partial pressure of arterial oxygen; **SARS CoV-2**, severe acute respiratory syndrome coronavirus 2; **SOFA**, sequential organ failure assessment.

## Results

The median age of the 2,440 study patients was 66 years; 61.6 percent were men, and 50.5 percent were Hispanic, Native American or African American. Age, gender, COPD, minimum mean arterial pressure, Glasgow Coma scale score, and $PaO_2/FiO_2$ ratio, maximum creatinine and bilirubin, receiving factor Xa inhibitors, days receiving non-invasive respiratory support and days receiving corticosteroids prior to intubation were significantly associated with the outcome variable. The validation cohort comprised 1,179 patients. C-TIME had the highest AUROC of 0.75 (95%CI 0.72–0.79), vs 0.67 (0.64–0.71) and 0.59 (0.55–0.62) for APACHE and SOFA, respectively ($Chi^2$ P<0.0001).

## Conclusions

C-TIME is the only mortality prediction score specifically developed and validated for COVID-19 patients who require mechanical ventilation. It has acceptable discriminant accuracy and goodness-of-fit to assist decision-making just prior to intubation. The C-TIME mortality prediction calculator can be freely accessed on-line at https://phoenixmed.arizona. edu/ctime.

## Introduction

The Coronavirus disease 2019 (COVID-19) pandemic raised concern that an overwhelming surge of critically-ill patients might require exclusion of patients with high predicted mortality from receiving mechanical ventilation [1]. The majority of COVID-19 ventilator triage policies surveyed in 2020 incorporated the Sequential Organ Failure Assessment (SOFA) score to predict mortality [2]. The SOFA score was originally designed to predict the mortality of sepsis patients based on assessment of the respiratory, renal, cardiovascular, hepatobiliary, coagulation and central nervous systems [3], and was externally validated in general ICU patient populations [4, 5]. However, a recent study using SOFA score data collected 48 hours prior to intubation in patients with COVID-19 pneumonia yielded a discriminant accuracy for mortality prediction of only 0.59 (95%CI: 0.55–0.63) [6]. Among other general ICU mortality scoring systems, the acute physiology and chronic health evaluation, version IVa (APACHE IVa) is notable, incorporating 145 variables and disease-specific regression models [7]. APACHE IVa has been shown to have superior discriminant accuracy compared to other general ICU mortality prediction models [8–10] and has been externally validated for COVID-19 patients [11], but it is based on variables obtained at the time of admission rather than at the time of intubation. Although many other scoring systems have been specifically developed to predict mortality in patients with COVID-19 [12–34], none focused on assessing the patient at the time of intubation, when patients, families and providers are forced to make critical decisions regarding life support. Informed consent for endotracheal intubation should include an objective discussion of prognosis, and the need for ventilator triage based on predicted mortality might yet arise in future regional covid hotspots. Therefore, the point in time when it becomes apparent that a patient with COVID-19 pneumonia is going to require mechanical intubation is arguably the most important time to determine their prognosis. Our aim was to develop a mortality prediction system we called C-TIME (Covid-19 Time of Intubation Mortality Evaluation) using variables typically available in the 48 hours before intubation, in order to inform consent, frame family expectations and assist end-of-life planning. Our secondary aims were to

validate C-TIME, determine its discriminant accuracy and calibration, and compare it to SOFA and APACHE IVa mortality prediction models.

## Materials and methods

### Study design

A retrospective cohort study, approved by the research determination committee of the University of Arizona IRB, with waived necessity for consent to use de-identified data, was set in 18 medical surgical ICUs in the Southwest United States between 6/1/2020 and 3/23/2021. June was chosen for cohort inception when preliminary results of the RECOVERY trial [35] were released, and administration of dexamethasone rapidly adopted in our study ICUs. We randomly assigned study patients to model-development and validation cohorts.

### Participants

Consecutive ICU patients were included based on the following eligibility criteria: ≥18 years of age; positive SARS-CoV 2 RNA by reverse transcriptase polymerase chain reaction or positive rapid antigen test; and undergoing endotracheal intubation ≥4 hours after admission. All patients were followed until hospital discharge or death.

### Variables and data sources

The main outcome variable was hospital mortality or discharge to hospice after terminal extubation–henceforth this combined outcome is referred to as "mortality". We chose candidate predictor variables to use in model development based on previous literature [12–34] and hypotheses generated by our clinical research team. We examined our clinical dataset and only included candidate predictor variables that were missing in less than 10% of study patients. We made an exception for the partial pressure of arterial oxygen/fraction of inspired oxygen ($PaO_2/FiO_2$) ratio, which we hypothesized would be a particularly important predictor [36]; therefore we planned a-priori to impute missing $PaO_2/FiO_2$ data (see statistics section below).

The following 25 candidate predictor variables, collected in the time period before intubation, were chosen to include in model development. Patient characteristics included: age, gender, body mass index (BMI), prior history of diabetes mellitus, hypertension, COPD, coronary artery disease, cancer or solid organ transplant. Physical examination findings included maximum temperature, lowest mean arterial pressure and lowest Glasgow Coma scale in the 48 hours prior to intubation. Laboratory variables included the highest concentration of creatinine and bilirubin, and the lowest platelet count and $PaO_2/FiO_2$ ratio in the 48 hours prior to intubation. Management variables comprised hospital days prior to intubation; hospital days receiving non-invasive respiratory support (high-flow nasal canula oxygen; continuous positive airway pressure or bilevel positive airway pressure) prior to intubation; hospital days receiving corticosteroids (dexamethasone, methylprednisolone or prednisone) prior to intubation; and administration of any of the following drugs: corticosteroids, therapeutic dose heparin/enoxaparin, oral Xa inhibitors, subcutaneous or intravenous insulin, or norepinephrine infusion. We also included intubation during surge conditions, defined as the time period(s) during which ≥ 400 ventilators (>5.5 ventilators per 100,000 population) were in use by COVID-19 patients in the state of Arizona where most of our study hospitals were located. By this criteria, surge conditions occurred in our ICUs in the summer (6/23/2020–8/7/2020) and winter (12/3/2020–2/14/2021) [37].

Variables needed to calculate the SOFA score were also extracted from the Cerner Millennium® electronic medical record, using the worst values in the 48 hours prior to intubation. SOFA variables include: PaO2, FiO2, use of invasive or non-invasive ventilatory support, lowest

MAP, use of intravenous vasopressors, GCS, platelet count, serum creatinine and bilirubin. These variables are used to assign a score of 0–4 to each of the corresponding organ systems– higher scores indicating worse organ function. The resulting cumulative SOFA score ranging from 0–24 determines predicted mortalities of 0–95% based on previous validation studies [3–5].

Data used to calculate the APACHE IVa predicted mortality were collected by direct electronic interface between Cerner Millennium® and Philips Healthcare Analytics. These included the worst physiological values occurring during the first ICU day, chronic health conditions and admission information. Predicted hospital mortality were provided by Philips Healthcare using proprietary APACHE IVa methodology (Cerner Corp. Kansas City, MO) [7].

## Study size

We calculated that a sample size of 2500 patients would allow analysis of 25 candidate predictor variables in our logistic regression. This was based on assumed mortality of 50%, providing 25 events for each predictor variable in both the model-development and validation cohorts.

## Statistical analysis

All study patients underwent randomization into one of the two cohorts. Missing $FiO_2$ values were imputed as the mean $FiO_2$ for all study patients for whom $FiO_2$ was known. Missing $PaO_2$ values were imputed as the mean $PaO_2$ of all study patients receiving the same $FiO_2$. The 25 candidate predictor variables were entered into backwards, step-wise, multiple logistic regression (MLR) using the model-development cohort, with mortality as the dependent outcome variable. We retained all variables that remained in the model at $P \leq 0.05$.

The MLR logistic equation was then applied to calculate predicted mortality for each patient in the validation cohort and this data was used to calculate area under the receiver operator curve (AUROC), Nagelkerke's pseudo $R^2$ and Hosmer-Lemeshow goodness-of-fit tests. AUROC of all patients for whom all three predicted mortalities (C-TIME, SOFA and APACHE IVa) were calculable were compared using the Chi-squared statistic. Calibration of each of the three models were compared using calibration belts [38]. We explored model performance in relation to the assumption that a predicted mortality of $\geq 75\%$ or $\geq 90\%$ might influence the decision whether or not to intubate. Therefore, we identified patient subgroups with $\geq 75\%$ and $\geq 90\%$ predicted mortality for each of the three models, and enumerated the *observed* mortality for each subgroup. This allowed calculation of the sensitivity of each model for mortality (the observed mortality in each subgroup divided by overall mortality) at the two cutoffs. The Wilson method was used to calculate 95% confidence intervals for single proportions. We used STATA® Version 17 (Statacorp, College Station, TX) for all statistical analyses.

## Sensitivity analysis

We calculated AUROCs for C-TIME and SOFA using only validation cohort patients for whom $FiO_2$ and $PaO_2$ were known (i.e. *excluding* patients with imputed values). These were compared to the AUROCs of our primary analysis by using z-tests on equality of proportions to test whether data imputation affected AUROC. Recalculation of APACHE IVa AUROC was not necessary because it did not incorporate imputed data.

## Results

Between 6/1/2020 and 3/23/2021, 18,431 patients with COVID-19 were admitted to study hospitals. Of these, 4,695 were admitted to the ICU and 2,440 were intubated $\geq 4$ hours after admission. Characteristics of these 2,440 study patients are presented in the Table 1. The

**Table 1. Clinical characteristics of 2440 study patients.**

| | Model Development Cohort (n = 1,221) | Validation Cohort (n = 1,219) |
|---|---|---|
| **Age in years, median (IQR)** | 66 (57–74) | 66 (56–75) |
| **Age in years, no. (%)** | | |
| 18–44 | 122 (10.0%) | 123 (10.1%) |
| 45–64 | 429 (35.1%) | 436 (35.8%) |
| 65–74 | 395 (32.3%) | 347 (28.5%) |
| 75–84 | 226 (18.5%) | 273 (22.4%) |
| >85 | 49 (4.0%) | 40 (3.3%) |
| **Male, no. (%)** | 740 (60.7%) | 762 (62.4%) |
| **Race/ethnicity, no. (%)*** | | |
| Non-Hispanic white | 542 (44.4%) | 549 (45.0%) |
| Hispanic | 481 (39.4%) | 462 (37.9%) |
| Native American | 88 (7.2%) | 94 (7.7%) |
| African American | 45 (3.7%) | 63 (5.2%) |
| Asian/Pacific Islander | 22 (1.8%) | 18 (1.5%) |
| Other/Multiple Race/Unknown | 43 (3.5%) | 33 (2.7%) |
| **Body Mass Index, median (IQR)** | 31.3 (27.2–37.1) | 31.8 (27.4–38.0) |
| **Admitted during surge, no. (%)** | 897(73.5%) | 893(73.3%) |
| **Medications, no. (%)** | | |
| Steroids | 1068 (87.5%) | 1,046 (85.8%) |
| Insulin | 655 (53.6%) | 680 (55.8%) |
| Therapeutic heparin/enoxaparin | 105 (8.6%) | 96 (7.9%) |
| Oral Xa inhibitors | 77 (6.3%) | 83 (6.8%) |
| Norepinephrine | 249 (20.4%) | 233 (19.1%) |
| **Comorbidities, no. (%)** | | |
| Diabetes | 726 (59.5%) | 730 (60.1%) |
| Hypertension | 929 (76.1%) | 922 (75.6%) |
| Coronary Artery Disease | 343 (28.1%) | 353 (29.0%) |
| COPD | 187 (15.3%) | 180 (14.8%) |
| Cancer | 117 (9.6%) | 126 (10.3%) |
| Solid organ transplant | 17 (1.4%) | 17 (1.4%) |
| **Physical examination, mean (SD)** | | |
| Minimum mean arterial pressure (mmHg) | 70.7 (61.7–80.3) | 71.0 (62.3–80.0) |
| Maximum temp (˚C.) | 99.0 (98.4–100.0) | 99.0 (98.4–99.9) |
| Minimum Glasgow Coma Scale score, median (IQR) | 15 (14–15) | 15 (14–15) |
| **Labs, median median (IQR)** | | |
| C-reactive protein, mg/L | 120.5 (62.4–194.8) | 126.6 (71.6–209.4) |
| Creatinine**, mg/dl | 0.93 (0.7–1.4) | 0.97 (0.7–1.5) |
| Bilirubin**, mg/dL | 0.6 (0.4–0.8) | 0.5 (0.4–0.8) |
| $PaO_2/FiO_2$ ratio** | 73.7 (58.0–79.0) | 73.7 (57.0–80.6) |
| Platelets**, K/mm$^3$ | 229 (163–303) | 223 (160–300) |
| **Pre-intubation hospital course, median (IQR)** | | |
| Hours from admission to intubation | 91.8 (31.4–213.1) | 86.5 (31.8–190.3) |
| Days on non-invasive respiratory support before intubation | 3.0 (1.0–7.0) | 3.0 (1.0–6.0) |
| Days receiving steroids before intubation | 4.0 (1.0–8.0) | 3.0 (1.0–8.0) |
| **Outcomes, no. (%)** | | |
| In-hospital death | 771(63.2%) | 789 (64.6%) |

*(Continued)*

**Table 1.** (Continued)

| | Model Development Cohort (n = 1,221) | Validation Cohort (n = 1,219) |
|---|---|---|
| Terminal extubation and discharge to hospice | 42 (3.6%) | 5 (0.4%) |
| Combined death/discharge to hospice | 813 (66.6%) | 794 (65.1%) |

*Race/ethnicity was as reported by the patient at time of admission.

**Variables incorporated into SOFA score.

median age was 66 years, 61.6 percent were men, and 50.5 percent were Hispanic, Native American or African American. Eighty-six percent of patients received corticosteroids. Eleven variables were significant in the final MLR model (see Table 2). The validation cohort comprised 1,219 patients of whom 1,179 had complete data for analysis by MLR. Observed mortality in the validation cohort was 65.1%.

C-TIME AUROC was 0.75 (95%CI 0.72–0.79), Nagelkerke's pseudo $R^2$ = 0.25, and the Hosmer-Lemeshow $Chi^2$ showed acceptable goodness-of-fit with P = 0.29. (Note: this P value >0.05 shows that there is no significant difference between predicted and observed mortalities in subgroups of cohort patients, i.e., good calibration).

Two-hundred seventeen of 1179 validation cohort patients did not meet criteria for APACHE IVa calculations, and the remaining 962 patients were included in our comparison between the three models. The median (Inter-quartile range) of predicted mortality from C-TIME, SOFA and APACHE were 0.71 (0.54–0.83), 0.18 (0.07–0.26) and 0.20 (0.09–0.38) respectively. C-TIME had the highest AUROC of 0.75 (95%CI: 0.72–0.79), vs 0.67 (0.64–0.71) and 0.59 (0.55–0.62) for APACHE and SOFA, respectively ($Chi^2$ P<0.0001) (see Fig 1). C-TIME was well calibrated (see Fig 2), with P = 0.215 [C-TIME predicted mortalities were not significantly different from observed mortalities]. APACHE and SOFA had poor overall calibration (see Figs 3 and 4), deviating significantly from observed mortality (P<0.001 for each). Calibration belt plots showed APACHE and SOFA were only acceptably calibrated when predicted mortality was ≥84% and >73% respectively, and post-hoc analysis revealed

**Table 2. Multiple logistic regression model with significant predictor variables for the outcome mortality in the model-development cohort.**

| Significant predictor variables in the C-TIME MLR model: | Odds ratio* (95% CI) | P value |
|---|---|---|
| Age (years) | 1.71 (1.47–1.98) | <0.001 |
| Male Gender | 1.41 (1.06–1.89) | 0.019 |
| COPD | 1.63 (1.07–2.49) | 0.024 |
| Minimum mean arterial pressure (mmHg) | 0.81 (0.70–0.93) | 0.004 |
| Minimum Glasgow Coma Scale score | 0.82 (0.70–0.95) | 0.008 |
| $PaO_2/FiO_2$ (mmHg) | 0.73 (0.62–0.86) | <0.001 |
| Maximum creatinine (mg/dl) | 1.17 (1.00–1.36) | 0.050 |
| Maximum bilirubin (mg/dl) | 1.55 (1.13–2.13) | 0.006 |
| Days receiving non-invasive respiratory support | 1.52 (1.08–2.13) | 0.017 |
| Days receiving corticosteroids | 1.43 (1.05–1.94) | 0.024 |
| Received oral Xa inhibitors | 2.37 (1.16–4.85) | 0.018 |

*Odds ratios are associated with a one standard deviation (SD) increment for continuous variables. Values used for SD: age: 13.7 years, MAP: 13.7 mmHg, $PaO_2/FiO_2$: 78.3 mmHg, creatinine: 1.9 mg/dl, bilirubin: 2.0 mg/dl, days receiving corticosteroids: 5 days; minimum Glasgow Coma Scale score: 3; days receiving non-invasive respiratory support before intubation: 5 days.

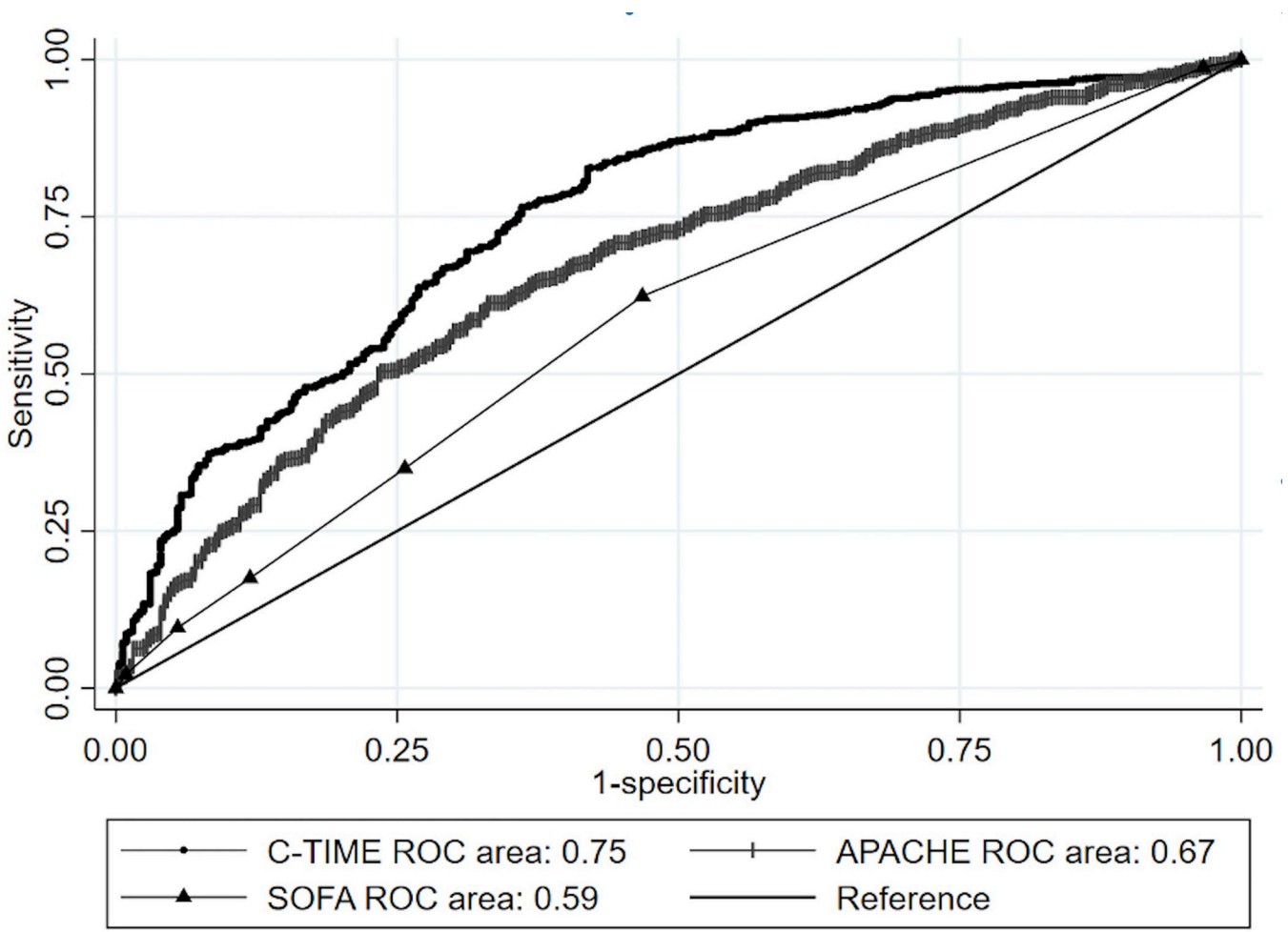

**Fig 1. Comparative AUROC of C-TIME, APACHE IVa, and SOFA mortality prediction systems.**

that mortality in that range was uncommonly predicted by either method. For instance, we noted 84% of patients had SOFA scores $\leq 9$ corresponding with <26% mortality.

C-TIME classified 486 patients as having $\geq 75\%$ mortality, of whom 399 died, yielding sensitivity of 50% (95%CI: 47–54%) at that cutoff. In comparison, APACHE and SOFA classified 46 and 120 patients as having $\geq 75\%$ mortality yielding sensitivities of 5%, (95%CI:4–7%) and 12% (95%CI: 9–14%) respectively. C-TIME classified 141 patients as having $\geq 90\%$ mortality, of whom 128 died, yielding sensitivity of 16% (95%CI: 14–19%) at that cutoff. In comparison, APACHE and SOFA classified 15 and 0 patients as having $\geq 90\%$ mortality, yielding sensitivities of 2% (95%CI:1–3%) and zero respectively.

## Sensitivity analysis in relationship to imputed data

Eighty percent of study patients for whom $FiO_2$ was recorded had an $FiO_2$ of 100%. $FiO_2$ was imputed to be 96% in 202/2440 patients (8.3%) with missing data. $PaO_2$ was imputed in 647/2440 patients (26.5%). Sensitivity analysis showed that C-TIME and SOFA AUROCs in the subset of 896 validation patients *without* imputed $PaO_2$ were 0.75 (CI 0.71–0.79) and 0.58 (CI 0.54–0.62) respectively–identical to AUROCs calculated for the full validation cohort.

## A.

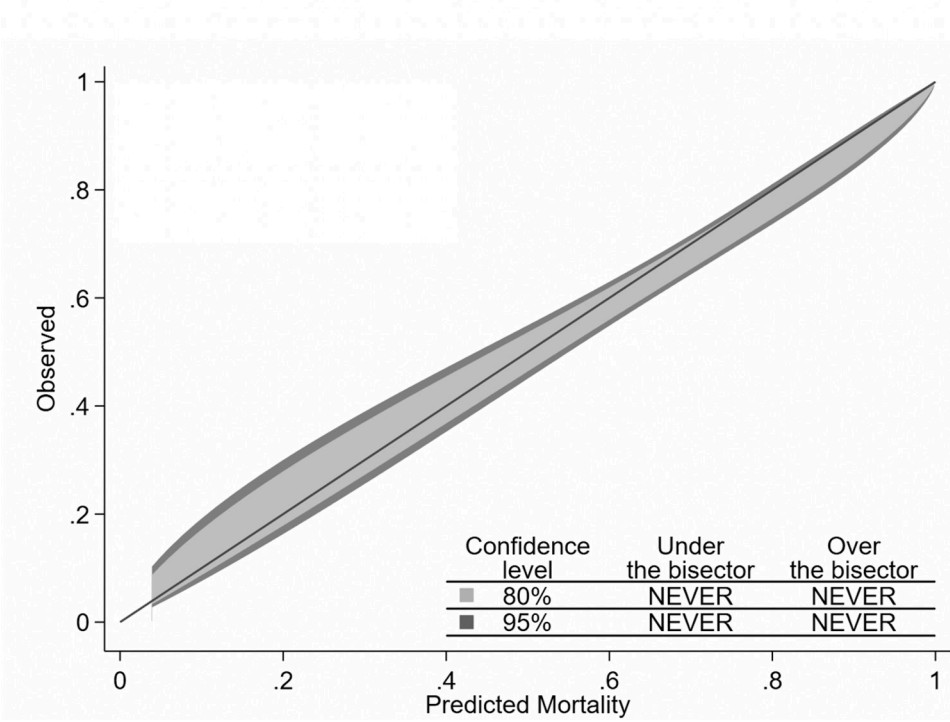

**Fig 2. Comparative calibration belt plots for C-TIME.**

## Discussion

The C-TIME mortality prediction model, based on eleven easily obtained clinical and laboratory variables, has better discriminant accuracy than APACHE IVa with 145 variables [7]. Furthermore, the C-TIME model has acceptable calibration and sensitivity in patients with high predicted mortality, in whom C-TIME may be helpful in making end-of-life decisions. Our study hospitals range from tertiary academic centers to community and critical access facilities serving a variety of persons from urban and rural communities with a wide diversity of racial/ ethnic backgrounds and socioeconomic status, enhancing the external generalizability of our findings.

Well over one hundred prognostic systems, including general ICU systems (such as SOFA and APACHE), and novel systems specifically developed for COVID-19 patients have already been published to predict clinical outcomes in patients with COVID-19 [12]. These vary by target patient population, predictor variables and outcomes of interest. To provide context for C-TIME, we reviewed comparable scoring systems that were developed and validated specifically for hospitalized COVID-19 patients and which incorporated commonly available clinical and laboratory predictor variables, and which reported AUROCs for in-hospital mortality [13–34].

Several features distinguish C-TIME from other validated COVID-19 mortality prediction systems we reviewed. 1) C-TIME is the only system that specifically evaluates patients with COVID-19 pneumonia just before they require mechanical ventilation. The discriminant accuracy of other prognostic models at this point in a patient's clinical course are unknown, due to spectrum effect (40). 2) The C-TIME study cohort had by far the highest reported mortality (65%) of any of the previous studies, as would be expected for *intubated* COVID-19

B.

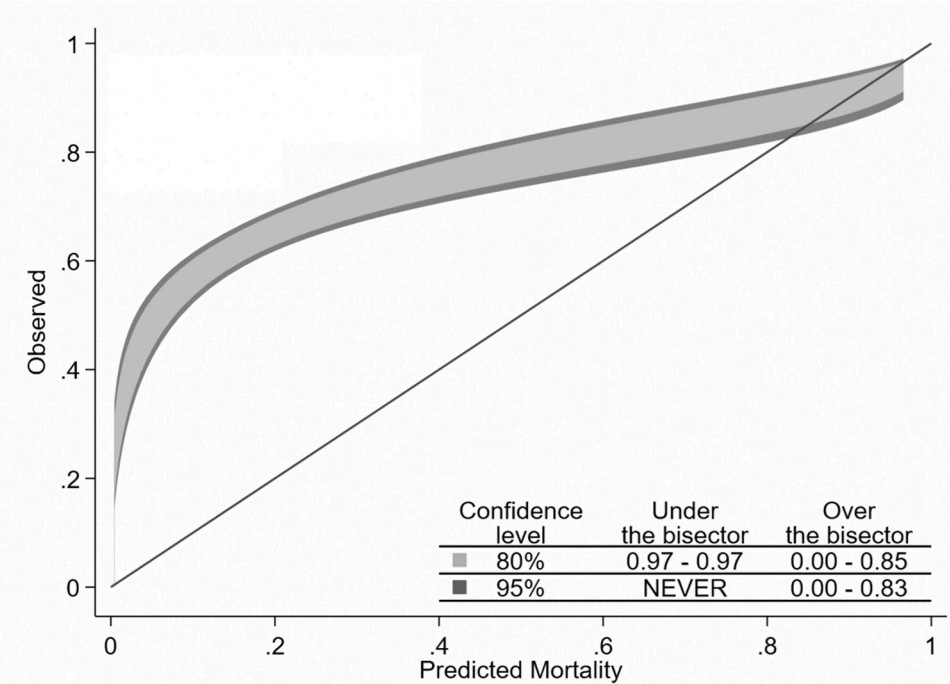

**Fig 3. Comparative calibration belt plots for APACHE IVa.**

patients [39]. The mortality of the study cohort has a strong influence on the operating characteristics of associated mortality prediction systems [40]–another reason why previously reported mortality prediction scores are likely not accurate if used at the time of intubation. 3) Other mortality prediction systems utilized study cohorts that included patients admitted prior to 6/2020, when preliminary results of RECOVERY were released. The inclusion of significant numbers of patients who did not receive corticosteroids could limit their generalizability in relationship to current practice patterns. Eighty-six percent of our study patients received corticosteroids before intubation. 4) C-TIME is the only model that incorporates treatment variables. Days receiving corticosteroids and days receiving non-invasive respiratory support prior to intubation were associated with mortality in our model-development and validation cohorts, and were also significantly associated with surge conditions (p = 0.0003 and 0.005, respectively). Surviving patients received a median of two days steroids and two days non-invasive respiratory support; non-survivors received a median of nine days steroids and eight days non-invasive respiratory support. A recent study showed that mortality increased significantly during the winter 2020 COVID-19 surge [41] however a meta-analysis concluded that delaying intubation does *not* influence mortality [42]. It is possible that the associations observed in our study might be due to prolonged efforts at non-invasive respiratory support and corticosteroid treatment of patients during surge conditions, selecting treatment non-responders for intubation.

One particular C-TIME variable deserves brief comment–the association of factor Xa inhibitors with *increased* mortality. Analysis of a sample of these patients showed that pre-existing atrial fibrillation was the indication for factor Xa inhibitors in 80%. It is possible that receiving

C.

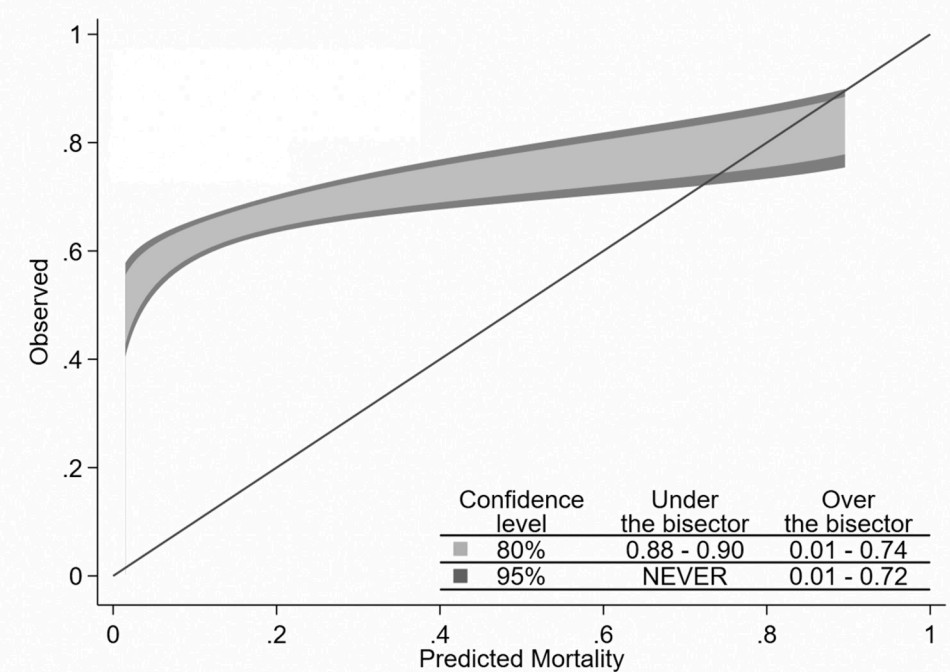

**Fig 4. Comparative calibration belt plots for SOFA.**

factor Xa inhibitors was a confounding variable representing pre-existing atrial fibrillation in our model.

Several other mortality prediction systems with acceptable operating characteristics are available for prognostication at the *time of admission* to the hospital, rather than at the time of intubation. The 4C score is supported by the largest study cohort and has an AUROC (0.77) similar to C-TIME [13]. We noted the highest AUROCs were reported for systems that prognosticated using variables from the time of admission in Hubei province early in the pandemic [16, 22, 26, 30, 32]. We feel these results are likely irreproducible outside the special circumstances under which they were reported. This contention is supported by a study using data from the Veterans Affairs Data Warehouse [19] that externally-validated two of these prediction scores [30, 32] and found much lower AUROCs than those originally reported: 0.68 vs. 0.91, and 0.72 vs. 0.94, respectively. This phenomenon was also demonstrated for the SOFA score, which achieved AUROCs of 0.89 (0.83–0.96) [43] and 0.99 (0.98–1.00) [44] in Hubei province early in the pandemic, versus 0.58 and 0.61 (0.53–0.70) in larger, more recent studies from the US and UK [6, 15].

The calibration belt plots in Fig 2 show that C-TIME has good fit across the entire range of mortality prediction, and that APACHE and SOFA have poor fit, underestimating mortality over much of the range. Calibration is better for APACHE and SOFA when predicted mortality is above 75%. However, APACHE and SOFA uncommonly predict mortality in this range; the upper limits of their IQR for predicted mortality are 38% and 26% respectively, despite the overall mortality of 65% in the validation cohort. This is consistent with findings of our prior study which showed that SOFA is likely to under-estimate mortality in covid patients at the time of intubation because many only have single organ system failure at that point [6]. Such a

patient, receiving 100% oxygen by bilevel positive airway pressure ventilation, with a resulting $PaO_2$ of 55 mmHg, would have a SOFA score of 4 indicating <10% mortality. The lack of sensitivity for APACHE and SOFA at cutoffs of 75% and 90% predicted mortality would severely limit their utility in identifying high risk patients less likely to benefit from intubation.

## Limitations of the study

Missing data was a major complication of our retrospective cohort design that limited us from including less-frequently-ordered predictor variables such as C-reactive protein, and led us to impute missing $PaO_2$ and $FiO_2$ data. Our sensitivity analysis showed that the later did not affect our AUROC estimates. Our EMR data source limited our ability to include variables not recorded as discrete data, such as COVID-19 vaccination status and pre-existing atrial fibrillation.

The discriminant accuracy achieved by C-TIME was modest, although similar to several other COVID-19 mortality prediction systems with AUROCs ranging 0.72–0.79 [15, 18, 25, 27, 29, 45]. We believe that it is inherently difficult to predict COVID-19 mortality at the time of intubation because such patients are relatively clinical homogeneous; most have life-threatening, single organ, respiratory failure (see Table 1) [3]. Low variation in predictor variables reduces discriminant accuracy. This could explain why APACHE IVa, which achieved AUROC of 0.88 in a large general ICU population [7], only yielded an AUROC of 0.66 in our study cohort.

C-TIME (and all other COVID-19 prognostic systems) are likely to lose discriminant accuracy over time, as factors influencing survival evolve. These factors might include advances in therapy and emergence of new viral strains. The aforementioned decline in discriminant accuracy for SOFA reported in Hubei vs the US and UK shows that discriminant accuracy reported in one historical setting may not be generalizable in later settings. Thus, any prognostic scoring system for COVID-19 will likely require repeated validation over time. We have begun the process of re-validating C-TIME using data collected during the Omicron surge.

## Conclusions

C-TIME is the only currently available mortality predictive score specifically developed and validated for COVID-19 patients who require intubation. It has acceptable discriminant accuracy and goodness-of-fit to assist informed consent for intubation and other end-of-life issues that occur specifically at this critical juncture in the patient's care. The C-TIME predicted mortality calculator can be accessed free on-line at: https://phoenixmed.arizona.edu/ctime

## Acknowledgments

**Guarantor statement:** Dr Raschke had full access to all of the data in the study and takes personal responsibility for the integrity of the data and the accuracy of the data analysis.

## Author Contributions

**Conceptualization:** Robert A. Raschke, Suresh Uppalapu, Nehan Sher, Steven C. Curry.

**Data curation:** Pooja Rangan, Sumit Agarwal.

**Formal analysis:** Robert A. Raschke, Pooja Rangan, Sumit Agarwal.

**Funding acquisition:** Steven C. Curry, C. William Heise.

**Investigation:** Robert A. Raschke, Sumit Agarwal.

**Methodology:** Robert A. Raschke, Pooja Rangan, Sumit Agarwal, Suresh Uppalapu, Nehan Sher.

**Project administration:** Steven C. Curry, C. William Heise.

**Resources:** Pooja Rangan, Sumit Agarwal, Steven C. Curry, C. William Heise.

**Software:** Pooja Rangan, Sumit Agarwal, C. William Heise.

**Supervision:** Robert A. Raschke, Steven C. Curry, C. William Heise.

**Validation:** Robert A. Raschke, Pooja Rangan.

**Visualization:** Pooja Rangan, Sumit Agarwal, Suresh Uppalapu, Steven C. Curry.

**Writing – original draft:** Robert A. Raschke.

**Writing – review & editing:** Robert A. Raschke, Pooja Rangan, Sumit Agarwal, Suresh Uppalapu, Nehan Sher, Steven C. Curry, C. William Heise.

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
