## [Decision Letter · Decision Letter 0]

11 Mar 2022

PONE-D-21-40997COVID-19 Time of Intubation Mortality Evaluation (C-TIME): A System for Predicting Mortality of Patients with COVID-19 Pneumonia at the Time They Require Mechanical Ventilation.PLOS ONE

Dear Dr. Raschke,

Thank you for submitting your manuscript to PLOS ONE. After careful consideration, we feel that it has merit but does not fully meet PLOS ONE’s publication criteria as it currently stands. Therefore, we invite you to submit a revised version of the manuscript that addresses the points raised during the review process.

You have developed and validated a promising mortality prediction system C-TIME in patients requiring intubation for COVID-19, which showed better discriminant accuracy compared with existed methods. The paper is well-written and you applied appropriate statistical methods. However, appropriate revisions and replies to reviewers’ comments are recommended before acceptance.

We look forward to receiving your revised manuscript.

Kind regards,

Yuyan Wang, Ph.D.

Academic Editor

PLOS ONE

Journal Requirements:

Additional Editor Comments:

1. The introduction part needs to be expanded by adding more backgrounds and making the paragraph more logical.

2. Missing problem for predictors needs more clarification and details.

3. Some revisions are suggested to make reported results more rigorous.

Reviewers' comments:

Reviewer's Responses to Questions

**Comments to the Author**

1. Is the manuscript technically sound, and do the data support the conclusions?

Reviewer #1: Yes

Reviewer #2: Yes

2. Has the statistical analysis been performed appropriately and rigorously? 

Reviewer #1: Yes

Reviewer #2: Yes

3. Have the authors made all data underlying the findings in their manuscript fully available?

Reviewer #1: Yes

Reviewer #2: Yes

4. Is the manuscript presented in an intelligible fashion and written in standard English?

Reviewer #1: Yes

Reviewer #2: Yes

5. Review Comments to the Author

Reviewer #1: Congratulations on the outstanding work. This paper suggests a simple and feasible method for predicting patient prognosis at time of intubation in Covid-19 related cases. Acessible variables have been used and a modest accuracy rate has been achieved, which makes the method reproducible and obtainable worldwide.

It is intriguing that anticoagulation was associated with worst prognosis and that anti Xa inhibitors were associated with increased mortality, specially beacause other studies have demonstrated the importance of anticoagulation in critical patients admitted due to Covid-19.

Reviewer #2: The authors developed and validated a mortality prediction system called CTIME (COVID-19 Time of Intubation Mortality Evaluation) for patients requiring intubation for COVID-19, which can be very useful in clinical application. I’m impressed by the well-written paper, especially the discussion part. Appropriate statistical methods are applied in this study, but there are substantial issues that need to be addressed for acceptance.

Major:

1. Introduction section lists many drawbacks of current methods, but it lacks logit and is little bit short. Suggest to expand the introduction content, for example, consider to give more background about APACHE Iva and SOFA

2. Does the missingness problem only exist in PaO2/FiO2? If not, what’s the missing proportion of other candidate predictors? How do you handle their missingness?

3. I’m confused about the final numbers of participants in both model development and validation? Could the author clarify the final validation sample size used for C-TIME, APACHE Iva and SOFA, respectively, and if the numbers are different, could you explain why?

Minor:

1. Abstract: when abbreviating the terms (APACHE Iva/ SOFA/ AUROC), suggest to use the full term the first time you use it, followed by abbreviations in parentheses.

2. Method, Study design: "We randomly split our cohort in half". Why splitting 2440 into 1221 and 1219 (Table 1), not 1220 and 1220?

3. Method, Data sources: suggest to briefly explain how APACHE Iva and SOFA calculates predicted hospital mortality.

4. Method, Study size: “We calculated that a sample size of 2500 patients would allow analysis of 25 candidate predictor variables in our logistic regression”. More details are needed. What are the parameters you used for this sample size estimation?

5. Method, Statistical Analysis: “AUROC compared using the Chi-squared statistic; Chi2 P<0.0001”, is this p-value for comparison among three methods or between two methods?

6. Table 1: suggest to add “Overall” column for all 2440 subjects description and “p-value” column for comparison between development and validation cohorts; “Admitted during surge/ Outcomes” missing “No. (%)”, and “Physical examination/ Pre-intubation hospital course” missing “median (IQR)”; “**” in “Creatinine/ Bilirubin/ratio/ Platelets”, forget any footnote? any missingness for variables described in Table 1?

7. Table 3: it seems predicted mortality probabilities from APACHE Iva and SOFA were much lower than predicted probabilities from C-TIME. It would be helpful if the author could give the distributions of the predicted probabilities of these three methods. (Not necessarily add in the results, but just to check the distribution and explain numbers in Table 3)

8. What’s the final sample size in sensitivity analysis?

6. PLOS authors have the option to publish the peer review history of their article (what does this mean?). If published, this will include your full peer review and any attached files.

Reviewer #1: **Yes: **Beatriz Martinelli Menezes Goncalves

Reviewer #2: No

---

## [Author Response · Author response to Decision Letter 0]

20 Apr 2022

To the Editors PLOS ONE and the reviewers,

Thank you for your thoughtful consideration of our manuscript and recommendations to improve it. We have extensively rewritten the paper to address your concerns and those of the reviewers. We added calibration curves, an important methodological criteria for critical appraisal of research involving prognostic models (BMJ 2020;369:m1328). This additional analysis was revealing and entailed replacement of a table with a three-part figure in the revised version of our manuscript. 

Editors comments:

1) Our manuscript meet’s PLOS ONE’s style requirements.

2) We are restricted by our Data Use Agreement (DUA) with Banner Health from sharing data without a request. Here is the applicable section from the DUA: 

“The Data will be used solely to conduct the Project and solely by Recipient Scientist and Recipient’s faculty, employees, fellows, students, and agents (“Recipient Personnel”) and Collaborator Personnel (as defined in Attachment 3) that have a need to use, or provide a service in respect of, the Data in connection with the Project and whose obligations of use are consistent with the terms of this Agreement (collectively, “Authorized Persons”).Except as authorized under this Agreement or otherwise required by law, Recipient agrees to retain control over the Data and shall not disclose, release, sell, rent, lease, loan, or otherwise grant access to the Data to any third party, except Authorized Persons, without the prior written consent of Provider. Recipient agrees to establish appropriate administrative, technical, and physical safeguards to prevent unauthorized use of or access to the Data and comply with any other special requirements relating to safeguarding of the Data.” 

We have designated a non-author individual with data access to be contacted for data requests: Kieran Richardson, Research Development Manager, University of Arizona College of Medicine-Phoenix (KieranR@arizona.edu). 

3) We have included our full ethics statement in the Methods section as requested. 

4) Our reference list is complete and correct. None have been retracted. 

5) The introduction was expanded and clarified.

6) The problem of missing data for predictor variables was clarified.

7) Revisions suggested to make reported results more rigorous have been made.

Response to Reviewers.

We greatly appreciate the reviewers comments.

Reviewer #1 did not request any revisions.

Reviewer #2 Major revision requests:

1. Introduction section lists many drawbacks of current methods, but it lacks logic and is little bit short. Suggest to expand the introduction content, for example, consider to give more background about APACHE Iva and SOFA. 

We expanded the introduction, improving the logic, and adding background regarding APACHE and SOFA. We added references currently numbered 4,5 and 8-11, to support this expansion of the introduction. 

2. Does the missingness problem only exist in PaO2/FiO2? If not, what’s the missing proportion of other candidate predictors? How do you handle their missingness?

We did not include esoteric predictor variables in our model development, but only variables which >90% of the patients in our study population had available in the 48 hours prior to intubation due to typical clinical practice. Multiple logistic regression only included patients with no missing data (except for P/F). In the validation cohort, 1179/1219 (96.7%) of patients had results for all predictor variables (except P/F) – conversely, 40 patients (3.3%) were missing at least one of the predictor variables and could not be included in MLR. 

The only exception to missing data was PaO2/FiO2. We provide the rationale for why we imputed missing P/F data in methods. Several of our study site hospitals used a protocol that encouraged use of oximetry and capnography instead of serial arterial blood gases in the monitoring of mechanically ventilated patients, and we were therefore missing P/F data on a surprisingly large minority of patients, but we felt PaO2 was too important of a variable to be left out of the model. Therefore missing P/F data were imputed and a sensitivity analysis performed which compared AUROC in patients with and without imputed P/F data, that showed essentially identical AUROCs whether or not imputed P/F data were used. 

3. I’m confused about the final numbers of participants in both model development and validation? Could the author clarify the final validation sample size used for C-TIME, APACHE Iva and SOFA, respectively, and if the numbers are different, could you explain why?

The validation cohort included 1219 patients of whom 1179 had complete data and were included in the MLR validation analysis. All 1179 patients had sufficient data to calculate SOFA scores, but 217/1179 (18.4%) were excluded from APACHE IVa analysis, provided by Philips Healthcare using proprietary APACHE IVa methodology. They have their own exclusion criteria independent from our study design, the most common of which are: ICU length of stay < four hours, missing data and transfer from another ICU. This left 962 patients for a fair comparison between C-TIME, SOFA and APACHE, as we thought comparing all 1179 patients with C-TIME and SOFA data to 962 patients with APACHE data would introduce bias. We clarified this in the revised manuscript. 

Minor:

1. Abstract: when abbreviating the terms (APACHE Iva/ SOFA/ AUROC), suggest to use the full term the first time you use it, followed by abbreviations in parentheses.

Done, although this increased the word count of the abstract.

2. Method, Study design: "We randomly split our cohort in half". Why splitting 2440 into 1221 and 1219 (Table 1), not 1220 and 1220?

We worded this poorly – the phrase “split our cohort in half” is inaccurate and has been clarified. Each patient’s randomization is independent of the randomization assignments of the other patients, so there is no guarantee that equal numbers of patients will be randomized to each group. We used block randomization to mitigate against large discrepancies, but small discrepancies like this are expected. 

3. Method, Data sources: suggest to briefly explain how APACHE Iva and SOFA calculates predicted hospital mortality.

This was added to the methods section. This entailed additional references currently numbered 4,5 and 8-11. 

4. Method, Study size: “We calculated that a sample size of 2500 patients would allow analysis of 25 candidate predictor variables in our logistic regression”. More details are needed. What are the parameters you used for this sample size estimation?

We have added the following statement: “ . . . based on the assumption of 50% mortality providing 25 events per predictor variable in each of the development and validation cohorts.” 

5. Method, Statistical Analysis: “AUROC compared using the Chi-squared statistic; Chi2 P<0.0001”, is this p-value for comparison among three methods or between two methods?

Between all three methods – the null hypothesis is that all three AUROCs are not significantly different that each other. 

6. Table 1: suggest to add “Overall” column for all 2440 subjects description and “p-value” column for comparison between development and validation cohorts; “Admitted during surge/ Outcomes” missing “No. (%)”, and “Physical examination/ Pre-intubation hospital course” missing “median (IQR)”; “**” in “Creatinine/ Bilirubin/ratio/ Platelets”, forget any footnote? any missingness for variables described in Table 1?

We made the minor changes to table 1 suggested above and added intended footnotes. 

Regarding statistical comparison of development and validation cohorts, we had originally included a column with P-values, but one of our coauthors (S Curry) argued strongly against it, on the basis that statistical comparisons are inappropriate in the absence of a research hypothesis. He suggested that we use the format typically used by the NEJM for comparing control and treatment groups, an example of which is pasted below. 

7. Table 3: it seems predicted mortality probabilities from APACHE Iva and SOFA were much lower than predicted probabilities from C-TIME. It would be helpful if the author could give the distributions of the predicted probabilities of these three methods. (Not necessarily add in the results, but just to check the distribution and explain numbers in Table 3)

This was a very helpful comment, and we learned a great deal trying to adequately respond to it. The median (Inter-quartile range) of predicted mortality from SOFA and APACHE are 0.18 (0.07-0.26) and 0.20 (0.09-0.38) respectively, which we didn’t appreciate in our former data analysis. We therefore reworked the manuscript quite a bit, eventually deciding to simplify, clarify and move the information previously in table 3 to the text of the results section and performing a formal treatment of calibration using the calibration belt technique of Nattino and Lemeshow (with new reference 39). This is added to the manuscript as figure 2, which graphically illustrates that APACHE and SOFA underpredict mortality over much of it’s range. This change is supported by a recent review on critical appraisal of research involving prognostic models, in which analysis of calibration is considered an important criteria (BMJ 2020;369:m1328). We also added a brief post-hoc analysis of the frequency of SOFA scores which showed that 83% of patients had SOFA scores <9 which correspond to predicted mortality 27%. This is quite illuminating since the overall observed mortality in the validation cohort was 65%. 

8. What’s the final sample size in sensitivity analysis?

This section was clarified – 896 patients in the validation cohort had non-imputed PaO2 and were used to recalculate AUROC. 

We hope this revision meets with your approval.

Respectfully,

Robert Raschke MD (and coauthors)

---

## [Editor Report · Decision Letter 1]

7 Jun 2022

COVID-19 Time of Intubation Mortality Evaluation (C-TIME): A System for Predicting Mortality of Patients with COVID-19 Pneumonia at the Time They Require Mechanical Ventilation.

PONE-D-21-40997R1

Dear Dr. Raschke,

We’re pleased to inform you that your manuscript has been judged scientifically suitable for publication and will be formally accepted for publication once it meets all outstanding technical requirements.

Kind regards,

Yuyan Wang, Ph.D.

Academic Editor

PLOS ONE

Additional Editor Comments (optional):

Thank you for addressing all comments and submitting your revised paper. Nice work!
---

## [Editor Report · Acceptance letter]

9 Jun 2022

PONE-D-21-40997R1 

COVID-19 Time of Intubation Mortality Evaluation (C-TIME):  A System for Predicting Mortality of Patients with COVID-19 Pneumonia at the Time They Require Mechanical Ventilation. 

Dear Dr. Raschke:

I'm pleased to inform you that your manuscript has been deemed suitable for publication in PLOS ONE. Congratulations! Your manuscript is now with our production department. 

Kind regards, 

on behalf of

Dr. Yuyan Wang 

Academic Editor

PLOS ONE